# Compositional Search of Stable Crystalline Structures in Multi-Component Alloys Using Generative Diffusion Models

## Abstract

Exploring the vast composition space of multi-component alloys presents a challenging task for both *ab initio* (first principles) and experimental methods due to the time-consuming procedures involved. This ultimately impedes the discovery of novel, stable materials that may display exceptional properties. Here, the Crystal Diffusion Variational Autoencoder (CDVAE) model is adapted to characterize the stable compositions of a well studied multi-component alloy, NiFeCr, with two distinct crystalline phases known to be stable across its compositional space. To this end, novel extensions to CDVAE were proposed, enhancing the model's ability to reconstruct configurations from their latent space within the test set by approximately 30% . A fact that increases a model's probability of discovering new materials when dealing with various crystalline structures. Afterwards, the new model is applied for materials generation, demonstrating excellent agreement in identifying stable configurations within the ternary phase space when compared to first principles data. Finally, a computationally efficient framework for inverse design is proposed, employing Molecular Dynamics (MD) simulations of multi-component alloys with reliable interatomic potentials, enabling the optimization of materials property across the phase space.

## 1 Introduction

Classical alloying strategies have already been known for millennia and remained relatively unchanged, i.e., small fraction of other material are added to metals to enhance their mechanical or thermal properties. For example, it has been discovered thousands of years ago that after adding a small weight percentage of other metals to silver, primarily copper, allows to produce sterling silver, which is a tougher version of pure silver. Up to present times, the same is true for steel where small amounts of carbon or chromium are added to iron in order to enhance its mechanical properties (George et al., 2019). Only 20 years ago, a novel alloying technique has emerged, proposing to mix multiple elements in relatively high concentrations, referred to as high-entropy alloys (HEAs)(Yeh et al., 2004; Cantor et al., 2004). This innovation has given rise to a new class of materials with exceptional properties, e.g., corrosion resistance or type-II superconductivity (Wu et al., 2016; Xiao et al., 2023). Consequently, there exists a multitude of potential HEAs, each differing in the number (such as binary, ternary, quaternary and quinary alloys) and/or type of constituent elements and their respective weight percentages, making them a fascinating subject for study. However, discovering stable HEAs with exceptional properties demands significant experimental efforts. Furthermore, achieving this through first-principles, specifically Density Functional Theory (DFT), requires a substantial computational effort. In light of these challenges, it is evident that novel tools are needed that would accelerate the search for materials within the HEA category. Here, generative deep learning models hold immense potential for efficient generation of new alloys.

In this paper, we take on this challenge and apply the Crystal Graph Diffusion Variational Encoder (CDVAE) model in a novel way for this problem. Usage of diffusion variational autoencoders for material generation has been proposed in Xie et al. (2022). Previously to Xie et al. (2022) variational autoencoders have been successfully used in molecular context Gebauer et al. (2019), which, however, are not periodic and thus allow for a less structured approach. While CDVAE have been successfully used to explore the composition space and discovering new structures in the context

of two-dimensional and superconducting materials (Wines et al., 2023; Lyngby & Thygesen, 2022), those solutions developed for these context do not deliver reasonable results for our case. Searching for HEAs requires us to consider the crystalline phase of the material in hand. More specifically, the atom types creating an HEA would be stable in different crystalline phases when the composition of the HEA is changed. Hence, here we need to take distinct approach for composition search within a HEA with specific constituent element types. For the specific example of NiFeCr, we first created a dataset including different compositions of it, including the binary alloys, with appropriate phases based on CALPHAD (Calculation of Phase Diagrams) method and using MD (Wu et al., 2017; Lee et al., 2001) which then was used for training the CDVAE model. It is a essential to consider big enough supercells in the dataset, i.e., structures of $3 \times 3 \times 3$ times a unitcell. This is needed so that a supercell contains enough atoms to represent possible percentage mixtures of materials. Next, we enhanced the CDVAE model by incorporating a fully-connected neural network for crystal phase classification - the resulting model is called P-CDVAE standing for Phase-CDVEA. This adjustment proved beneficial in considerably improving the denoising task's performance and as a result generation of materials with correct crystalline phase.

Afterwards, the model was used for exploring the ternary composition space and generating new materials. The stability of the newly generated materials were assessed by calculating their formation energies using DFT, which was with great agreement compared to other first principle methods aimed at the same study (Wróbel et al., 2015). Advancing one step beyond, we engineered a computationally efficient workflow aimed at optimizing material properties by incorporating feedback from MD. This enables a physics-informed inverse design methodology, provided that the interatomic potential being utilized accurately estimates the property under optimization.

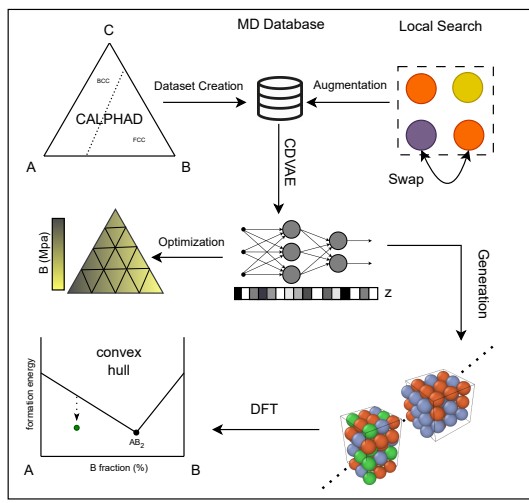

Figure 1: Composition search workflow.

Ultimately, we developed a local-search framework that, in essence, seeks the optimal property within a given configuration of specific composition by manipulating atomic ordering and utilizing feedback from MD simulations. This method was employed for augmentation of the initial training dataset by incorporating the optimized bulk modulus for each data point in the test set identifying the ultimate optimized configuration with regards to stability and mechanical properties. In summary, our primary contributions to this study are as follows:

- We generated a dataset for ternary NiFeCr, encompassing various atomic orderings of a given composition and the correct structural phases based on CALPHAD.
- We improved the denoising performance of the CDVAE by ceating P-CDVAE ,i.e., a new phase aware variational autoencoder model.
- We developed an optimization workflow that involves obtaining feedback from Molecular Dynamics (MD).
- We also integrated a local search method to augment the training dataset and determine the final structure of optimized composition within the phase space.

## 2 RELATED WORK

**Geometric graph neural networks (GNNs) for molecular graphs.** In recent years, several Graph Neural Networks (GNNs) have been developed with a specific emphasis on processing molecular graphs. These models employ graph geometry, representing atoms as nodes and their interatomic

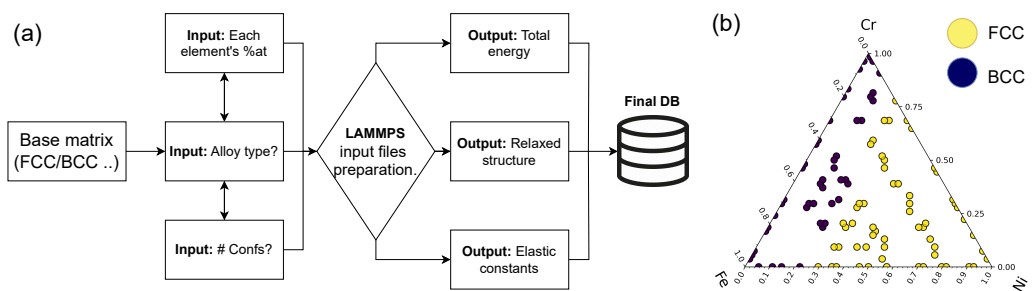

Figure 2: **(a)** Dataset creation workflow. **(b)**Training dataset created based on CALPHAD calculations and experimental data (Wu et al., 2017)

distances as edges, to generate various types of embedings. For instance, some models focus on edge lengths, such as Schnet (Schütt et al., 2017) and Physnet Unke & Meuwly (2019), while others consider both edge lengths and angles between them, e.g. DimeNet (Gasteiger et al., 2022b), GemNet-T (Gasteiger et al., 2022a), and AliGNN (Choudhary & DeCost, 2021). In its advanced version, GemNet-Q (Gasteiger et al., 2022a) goes a step further by incorporating torsion angles, which correspond to triplets of consecutive edges, to create positional embeddings. In this work we are using DimeNet as encoder and GemNet as decoder.

**Diffusion models for chemical structures** Generative models and have been applied to the materials science context extensively (Hoffmann et al., 2019; Long et al., 2021; Ren et al., 2022; Zhao et al., 2021). Diffusion autoencoders, however, are new class of generative models that have shown great performance at generating structured data by iteratively refining them (Dhariwal & Nichol, 2021; Cai et al., 2020; Shi et al., 2021). They manage various degrees of noise by optimizing a noise conditional score function, corresponding to the distortion of data, e.g. through the physics-inspired Langevin dynamics Song & Ermon (2019). The interpretations of score functions may vary, depending on the specifics of the problem and experimental decisions: a likelihood-based approach utilizes the probability distribution modeled by the network to maximize the likelihood of denoised data; for modeling molecular graphs, the formulation of energy-based diffusion is the most natural. CDVAE is an energy-based diffusion model that employs invariant representations of geometric GNNs and periodicity to model crystal graphs. However, there is a variety of approaches to modeling chemical entities: E(3) Equivariant Diffusion Model employs equivariant representations and diffusion based on the notion of likelihood (Hoogeboom et al., 2022).

**Quantum mechanical material discovery.** The quest for stable materials with exceptional properties poses a significant challenge when exclusively relying on quantum mechanical methods (Oganov et al., 2019). Efforts have been made to tackle this challenge through various approaches, including evolutionary algorithms(Wang et al., 2012; Glass et al., 2006), random sampling(Pickard & Needs, 2011), and the exploration of substitutional alloying within already established stable materials (Hautier et al., 2011). Among these methods, cluster expansion (CE) (Wu et al., 2016) techniques utilize quantum mechanical calculations to identify ground state stable structures in medium-entropy alloys (MEAs), and they are typically applied to alloys containing up to three components. However, it's worth noting that these methods may struggle with multi-component complex materials and the presence of multiple crystalline phases. In contrast, generative models offer a faster and more efficient approach when exploring the vast compositional space of materials.

## 3 METHODS

### 3.1 DFT AND MD CALCULATIONS

**Dataset creation with MD.** The training dataset is created for $3 \times 3 \times 3$ Face Centred Cubic (FCC) and Body Centred Cubic (BCC) super-cells of NiFeCr, NiFe, NiCr and only BCC FeCr in a range of compositions, validated to be stable based on modified embedded-atom Model (MEAM) interatomic potentials developed by Wu et al. (2017) and Lee et al. (2001). The mentioned MEAM potentials were validated by experimental data and CALPHAD (Calculation of Phase Diagrams) calculations

(for detailed information on the potentials see the references). Apart from the specific compositions studied in Wu et al. (2017), we added very small noise to the compositions of stable structures (less than 2 atomic percent (%at)), such that the final trained model has seen more data. The datset creation workflow is presented as a diagram in Figure.2 and is available as in (*Git repo will be added after the review*). The dataset creation starts by creating a crystal matrix with only one atom type. The structure phase (BCC or FCC in our case) of the initial matrix is selected based on the composition of the atoms as shown in Figure 2. Afterwards, the target structure with the specific composition is created by random substitution of second and third atom types with the first atom type in the matrix. In order to take into account the randomness of the target composition, 20 random configurations of a given composition is added to the dataset. LAMMPS (Thompson et al., 2022) is used for the molecular dynamics (MD) calculation of the formation energies and the elastic constants. The mentioned properties are calculated after conjugate gradient (CG) relaxation of the system at temperature T=0K.

**DFT validation.** The DFT spin-polarized calculations were performed with the `Vasp` package (Kresse & Furthmüller, 1996; Kresse & Hafner, 1993; Kresse & Furthmüller, 1996), using PAW PBE exchange-correlation functional (Blöchl, 1994; Kresse & Joubert, 1999) and an intial magnetic moment of NIONS$\times$5.0 (NIONS is the number of atoms in the cell). The Brillouin zone was sampled using a maximum $kpoint$ spacing of $0.5$ $\mathring{A}^{-1}$ on $\Gamma$-centred Monkhorst-Pack grids (Monkhorst & Pack, 1976) and plane-wave cutoff energy of $520$ $eV$ . Smearing with spreading of $0.05$ $eV$ was introduced within the Methfessel-Paxton method (Methfessel & Paxton, 1989) to help convergence.

## 3.2 CDVAE MODEL

**CDAVE**. To explore the ternary alloy composition space of the material being studied in this work (NiFeCr), we employed CDVAE (Crystal Diffusion Variational Autoencoder) developed by Xie et al. (2022). Diffusion autoencoders are trained in a self-supervised manner, by removing the artificially induced noise of varying magnitudes from the data. The model's ability to reverse various degrees of distortion is key for the generation, during which the model, prompted with a latent variable, iteratively refines the signal to obtain a correct example. CDVAE utilizes a physics-inspired method of annealed Langevin dynamics in the sampling process - after each iteration of querying the model, Gaussian noise is added to the data in order to enhance the exploration beyond the local optima. As the magnitude of the noise gradually decreases, the sampling approaches convergence (Song & Ermon, 2019). CDVAE consists of two geometric graph neural networks: DimeNet (Gasteiger et al., 2022b) as the encoder and GemNet (Gasteiger et al., 2022a) as the decoder. The model is equipped with a couple of fully-connected neural networks that utilize the latent vector to predict a certain parameters of the structure - those include number of atoms, elemental composition, the lengths of lattice vectors and the angles between them. Those are necessary to provide the graph input for GemNet: the atom types are at first chosen randomly from the probability ditribution based on the assessed elemental composition, and the atom coordinates are chosen from uniform distribution. GemNet is then used to output the score funtion for energy-based optimization, while also updating the previous prediction of atom types. A fully-connected network can also be trained to predict the properties of the output structure from the latent vector - the presence of such module enables the gradient optimization of properties in the latent space.

**P-CDVAE**. To determine the relative positions between adjacent atoms, one must take the crystal phase into consideration. Depending on the atomic composition, NiFeCr alloys form either FCC or BCC crystals, depending on the overall composition (Figure.2.(b)). To make the model correctly denoise the structures of either type, we equip it with a fully-connected network that predicts the crystal phase as from the latent vector and passes this information to representations of individual atoms before denoising is conducted. We refer to the model with this adjustment as P-CDVAE.

## 3.3 PROPERTY OPTIMIZATION

One of the potential applications of P-CDVAE is the generation of structures with desired properties. In addition to structure parameters, such as lattice, the number of atoms, and elementary composition, physical properties can be predicted and optimized using gradient descent. The CDVAE model is equipped with a fully-connected layer, which predicts the target feature among other structure-level parameters, prior to Langevin dynamics. During the optimization process, the embedding is

adjusted to maximize the property with Adam algorithm (Kingma & Ba, 2015), while the model's parameters are frozen. We use CDVAE to optimize the bulk elastic modulus of NiFeCr alloys. Each example in the test set is first encoded, and the encoding is passed to the fully-connected module that was trained to predict that property of the structure. We use MD simulations to assess the correctness of produced structures and calculate their bulk moduli.

**Local search for data augmentation.** The dataset is composed of structures obtained in MD simulations with randomly initialized atom positions. Mechanical properties of alloys are largely affected by chemical short range ordering/dis-ordering of atoms (Zhang et al., 2020; Naghdi et al., 2023). We hypothesize that the short-range configurations that are characterized by the highest bulk modulus consist of local structures that can't easily be obtained randomly. To provide the model with such examples, we extended the dataset by structures crafted with the use of local search. The operation used was tranposition of a pair of atoms. Under this operation, the atomic composition of the structure remains constant and it is ensured that the phase of the crystal doesn't change. After each operation, the structure is relaxed with MD. A crystal supercell S is characterized by the following parameters: an array of atom coordinates C, an array of atomic numbers Z, and three lattice vectors V. Modified structures are produced by swapping two values in Z. MD is then used to resolve the remaining parameters and estimate the bulk modulus of the emergent structure.

---

**Algorithm 1 :** Optimization of atomic structures

---

1: **Inputs:**
2: $S = \left\{ \mathbf{C} \in \mathbb{R}^{n_a \times 3}, \mathbf{Z} \in \mathbb{Z}^{n_a}, \mathbf{V} \in \mathbb{R}^3, \right\}$
3: $K \in \mathbb{R}, n_{\text{steps}} \in \mathbb{Z}, n_{\text{samples}} \in \mathbb{Z}$
4: **Definitions:**
5: $K_{MD}(S)$ denotes $K$ estimated by MD on $S$
6: **Algorithm:**
7: $K_{\text{best}} \leftarrow K$
8: $S_{\text{best}} \leftarrow S$
9: **for** $step = 1, \ldots, n_{\text{steps}}$ **do**
10:     $\mathbf{t} \leftarrow$ all pairs $(i, j)$ s.t. $Z_i \neq Z_j$
11:     $\mathbf{t}^* \leftarrow$ random subset of $\mathbf{t}$ with $n_{\text{samples}}$ elements
12:     $K_{\text{step}} \leftarrow K_{\text{best}}$
13:     $S_{\text{step}} \leftarrow S_{\text{best}}$
14:     **for** each $(i, j) \in \mathbf{t}^*$ **do**
15:         $\mathbf{C}^* \leftarrow \begin{cases} C_i^* = C_j \\ C_j^* = C_i \\ C_k^* = C_k \text{ for } i \neq k \neq j \end{cases}$
16:         $S^* \leftarrow \{\mathbf{C}^*, \mathbf{Z}, \mathbf{V}\}$
17:         **if** $K_{MD}(S^*) > K_{\text{step}}$ **then**
18:             $K_{\text{step}} \leftarrow$ K_MD$(S^*)$
19:             $S_{\text{step}} \leftarrow S^*$
20:     $K_{\text{best}} \leftarrow K_{\text{step}}$
21:     $S_{\text{best}} \leftarrow S_{\text{step}}$
22: **return** $K_{\text{best}}, S_{\text{best}}$

---

During each step of the procedure, a random subset of all possible transpositions is explored and the one that yields the highest improvement of the property is applied. The pseudocode of the procedure is shown in Algorithm 1.

## 3.4 RECONSTRUCTION SCORES

In order to cover the random ordering of the atoms in a specific point in the composition space, we need to chose relatively large examples that have freedom to form arbitrary permutations of all the atoms. However, we have observed that perfect reconstruction of the entire structure is unlikely. Therefore, we use Behler-Parinello (BP) (Behler & Parrinello, 2007) vectors, which are embeddings of local geometry for each atom , describing its local environment, to measure the similarity between the ground truth and the reconstructed configurations. In this work, PANNA: Properties from Artificial Neural Network Architectures (Lot et al., 2020), is used as an external package for calculation of the modified version of Behler–Parrinello (mBP) descriptors (Smith et al., 2017). The mBP representation generates a fixed-size vector, G[s], the "G-vector", for each atom in each configuration of the dataset. Each G-vector describes the environment of the corresponding atom of the configuration to which it belongs, up to a cutoff radius $R_c$. In terms of the distances $R_{ij}$ and $R_{ik}$ from the atom $i$ to its neighbors $j$ and $k$ and the angles between the neighbors $\theta_{jik}$. The radial and angular G-vectors are given by:

$$G_i^{rad}[s] = \sum_{i \neq j} e^{-\eta(R_{ij} - R_s)^2} f_c(R_{ij}) \tag{1}$$

$$G_i^{ang}[s] = 2^{1-\zeta} \sum_{j,k \neq i} [1 + cos(\theta_{ijk} - \theta_s)]^{\zeta}$$
$$\times e^{-\eta[\frac{1}{2}(R_{ij}+R_{ik})-R_s]^2} f_c(R_{ij})f_c(R_{ik}) \tag{2}$$

where the smooth cutoff function (which includes the cutoff radius $R_c$) is given by:

$$f_c(R_{ij}) = \begin{cases} \frac{1}{2}\left[\cos\left(\frac{\pi R_{ij}}{R_c}\right) + 1\right], & R_{ij} \leq R_c \\ 0, & R_{ij} > R_c \end{cases} \tag{3}$$

and $\eta$, $\zeta$, $\theta_s$ and $R_s$ are parameters, different for the radial and angular parts. The choice of the cutoff value is made so that it covers up to three nearest neighbours of the center atom in the BCC NiFeCr, which has an average lattice constant of $a = 3$ Å, and thus the third nearest neighbour's distance is $a \times \sqrt{2} = 4.24$ Å. This automatically covers up to the 4th nearest neighbour of the FCC crystal. We use Euclidean distance between G-vectors of two atoms as a measure of their dissimilarity. To assess the overall dissimilarity of two structures, we compute the minimum assignment between the G-vectors of their respective atoms. Then the average of the distances between all atom pairs in such assignment is taken, and we refer to that value as "G-distance" between a pair of structures.

## 4 EXPERIMENTS

### 4.1 LOCAL SEARCH

In order to improve the model's capability to take advantage of short-range order in the optimization, we created additional example with the use of composition-preserving local search. The procedure was conducted on 10% of the training, validation and test set. We observe that, as the procedure progresses, each iteration of the algorithm yields diminishing returns. We stop the search after a fixed amount of iterations, so as not to overfit to the Molecular Dynamics simulation used in the process. Figure. 3 shows the changes of bulk elastic modulus of the augmented portion of data during the local search optimization. We added training and validation structures optimized for 20 iterations to their respective dataset when training the augmented model (Aug). Each iteration involved generating 20 samples by swapping a pair of atoms.

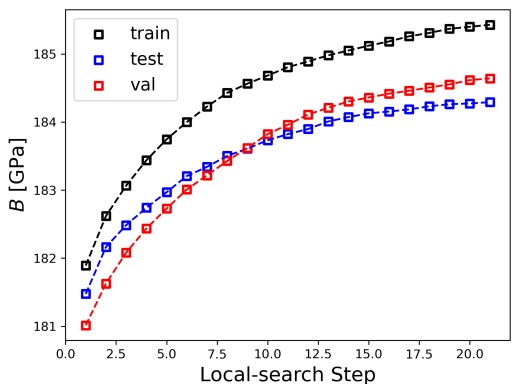

Figure 3: The mean values of bulk elastic modulus of a fraction of data during consequent steps of local search.

### 4.2 RECONSTRUCTION

We evaluate the model's capability to encode and reconstruct the examples from the test set. This task provides valuable insights into the extent to which the model can accurately restore both the atoms' chemical ordering (CO) and the crystal phase (CP) of the structures within the test set. We compute the G-vectors that provide representation of both geometric features and atom types to

Table 1: The reconstruction scores of models on the test set (without augmented examples)

| Model | Dataset | CO G-distance | Improv | CP G-distance | Improv | Validity |
|-------|---------|---------------|--------|---------------|--------|----------|
| CDVAE | Test | 0.3618 | - | 0.0984 | - | 92.0% |
| P-CDVAE | Test | **0.3513** | **2.9%** | 0.0691 | 29.37% | 93.9% |
| Aug-P-CDVAE | Test | 0.3521 | 2.7% | **0.0560** | **43.09%** | 97.6% |

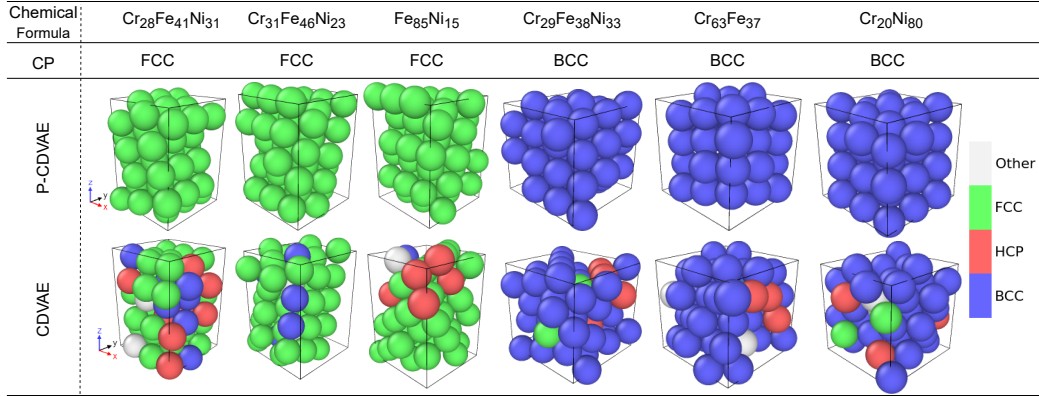

Figure 4: Reconstructed examples from P-CDVAE and CDVAE models, with atoms color coded by their corresponding crystal structure.

obtain G-distances between the ground truth and reconstructed structures. We define two separate metrics: Chemical Ordering (CO) G-distance, calculated from G-vectors as defined by (Behler & Parrinello, 2007), as well as crystal phase (CP) G-distance, calculated from modified G-vectors, in which atom types are disregarded - all nodes are treated like atoms of the same element.

**Results.** Table 1 shows the averaged reconstruction CO/CP G-distance scores of models on the randomly initialized test set. The P-CDVAE model has improved by about 30% compared to CDAVE. Evidently, the reconstruction scores are better for the augmented dataset as the model has seen more datapoints compared to P-CDVAE. To further analyse the reconstructed materials, polyhedral template matching algorithm (Larsen et al., 2016) from Ovito package (Stukowski, 2010) was employed for CP characterization. Figure. 4 depicts some examples in which P-CDVAE performed better in terms of reconstruction of the materials from their latent space. The atoms in Figure. 4 are color coded in terms of their lattice crystal structure. Obviously, P-CDVAE performed better on the depicted examples, as there are Hexagonal Close Packed (HCP) and unrecognized lattice crystal structures identified in all the examples, and also there are FCC structures in materials with BCC ground truth CP and vise the versa.

## 4.3 MATERIAL GENERATION

A significant motivation for utilizing generative models in materials discovery arises from the imperative to explore the vast composition space of materials and uncover the ones that were previously absent from the dataset. The success of this task depends on various factors, including the performance of the generative model framework, among others. As a result, we conducted an evaluation of both CDVAE and P-CDVAE in the reconstruction and generation task to determine their respective performance, ultimately choosing generated materials from P-CDVAE for subsequent DFT validation due to its enhanced capability in denoising the structures within their ground truth crystalline phase. To do so, the CDVAE and P-CDVAE models have been queried with latent vectors from multidimensional Gaussian distribution to generate 500 structures for each example. Figure.5.(a-b) show the CDAVE and P-CDVAE performance on denoising the newly generated materials into the correct CP, defined by the black line in the figures. The CP is determined by Ovito package in this section. Qualitatively, the P-CDAVE model is performing better in terms of correct CP, which is in correspondence with the reconstruction scores in the previous section. Choosing the P-CDVAE model as the "better" model, we show the distribution of the 500 generated configurations against the points in the dataset in Figure.5.(c), which shows that the vast part of the ternary phase diagram is covered. Afterwards, we did spin-polarized DFT calculations and based on the formation energy results, we found great agreement with the cluster expansion method (CE) in terms of the stability of the materials for FeNiCr (Wróbel et al., 2015). This benchmarks the method designed in this work promising for study of higher than three component alloys.

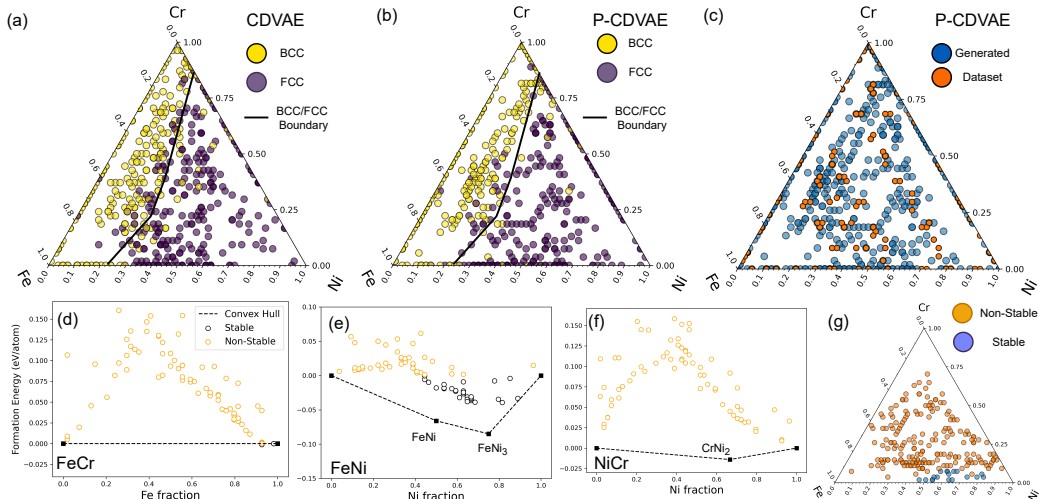

Figure 5: Material generation performance. **(a-b)** summarise both CDVAE and P-CDVAE models' ability to denoise newly generated structures to the *ground truth* CP, determined by CALPHAD calculations shown in Figure 2. **(c)** demonstrates the distribution of the newly generated naterials w.r.t the points covered in the initial dataset. In **(d-e)** the stable materials, validated in DFT, are shown for all the binary materials and corresponding ternary plot is presented in **(g)**.

## 4.4 OPTIMIZATION

The property maximization experiment was conducted by encoding the 327 structures from the test set and optimizing the encoding to maximize the bulk modulus. The encodings were used to sample new structures at several stages of optimization. MD was used to predict their bulk moduli and rule out the incorrectly denoised structures - we require that, after the simulation converges, per-atom energy is not higher than the threshold set by the highest per-atom energy found in the training data. Additionally, we require that the output of the neural model is already close to convergence - total energy of the structure can be at most 10 eV above the final threshold. As encodings are optimized with a simple goal to maximize the prediction of a fully-connected network, the moment when most of them cease to translate into structures that fit these criteria marks the end of the experiment. Table 2 shows the amount of correctly decoded structures, as well as the values of elastic moduli present in the population at consecutive stages of optimization.

Table 2: The results of optimization with P-CDVAE and Aug-P-CDVAE model. The latent vectors were used for sampling after fixed numbers of steps. Mean property in the optimized population of generated crystals, mean property of 10 best examples and number of structures that fit the energy criteria are presented.

| Model | Metric | Initial data | Step | | | | |
|---|---|---|---|---|---|---|---|
| | | | 200 | 500 | 1000 | 1500 | 2500 |
| P-CDVAE | Mean all [GPa] | 181.7 | 181.4 | 181,7 | 181,1 | 182.8 | 185.8 |
| | Top 10 mean [GPa] | 192.0 | 193.1 | 193.3 | 193.8 | 193.8 | 193.3 |
| | Correct | 327 | 237 | 225 | 178 | 129 | 45 |
| Aug-P-CDVAE | Mean all [GPa] | 181.7 | 181.7 | 182.4 | 185.4 | 189.8 | 193.0 |
| | Top 10 mean [GPa] | 192.0 | 192.9 | 193.0 | 194.4 | 195.0 | 196.5 |
| | Correct | 327 | 265 | 247 | 199 | 186 | 124 |

**Results**. P-CDVAE, trained on the data initialized from uniform distribution, showed little capability to optimize the bulk elastic modulus - the increase in the mean value in late stages of optimization is

not reliable due to the invalidity of most datapoints that the mean initially comprised. It is suspected that this improvement was caused mostly by the optimization of composition, which takes over in later stages of the process. The augmented model, however was capable of improving the examples with most beneficial compositions by several GPa, a value similar to that achieved by the augmentation. The information about beneficial local structures allowed it to produce the best structure of bulk modulus 197.26 GPa, a value slightly higher than the highest one obtained during local search - 196.58 GPa. Despite the simplistic setting of local search, the neural model possesses substantial edge in terms of computational efficiency. The exploration of different compositions during the optimization is discussed in the appendix A.

## 5 CONCLUSIONS

In this work, a computationally efficient pipeline for generating crystal structures has been proposed. The utilization of interatomic potentials for structures composed of nickel, iron and chromium, provided by Wu et al. (2017) and Lee et al. (2001), allowed for quick generation of training data with molecular dynamics (MD) simulations. The model yielded mostly valid configurations and posterior analysis showed that stable structures are among the generated examples. In the optimization process, the feats of MD were used once again to ensure the correctness of generated configurations and provide a preliminary assessment of the value of the optimized feature.

The reconstruction task for CDVAE model was not faithful enough to feasibly match the reconstructed examples to their ground truth counterparts with the StructureMatcher implemented in Pymatgen library Ong et al. (2013). A possible reason for that is a relatively large number of atoms in a single supercell. A similar observation was made by the authors of CDVAE - as the model encodes local structural patterns, it may likely come up with a different global structure. This phenomenon is possibly even more prevalent in our setup, with larger, 54-atom structures. Thus, we created P-CDVAE, which is a phase aware variational autoencoder model that improved this task's score by approximately 30 %.

The use of data produced with high-quality interatomic potentials can greatly accelerate the generation of crystal structures in comparison to creating training data with DFT. A clear liability of this approach is the risk of generating fewer stable structures, which may result in sunken cost during the posterior DFT analysis of the output structures. The proposed machine learning pipeline could be enhanced by another computationally efficient tool, for example a pretrained model for prior assessment of structures' stability (Wines et al., 2023).

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
