# A  OPTIMIZATION EXPERIMENTS

We assess the models' capability to find new maxima of bulk elastic modulus across the composition space. To indicate the relation between elemental composition and the property, we find the highest bulk modulus per each atomic composition present in the test set, which is the initial point of optimization. In the figure 1 we show the highest per-composition values, interpolated by triangulation (a). Those interpolated values can be improved either by finding new optimum for a given composition, or finding structures in unexplored regions with better properties than interpolation would indicate. We show interpolation plots, with input data consisting of test examples, enriched with structures yielded by the models (b for P-CDVAE, c for Aug P-CDVAE). We then show the improvement upon the baseline interpolation (d for P-CDVAE, e for Aug P-CDVAE). Ultimately, we show the improvements achieved by Aug-P-CDVAE w.r.t. the more optimized augmented dataset it used in training instead (f). The compositions for which a structure exists are marked by black dots.

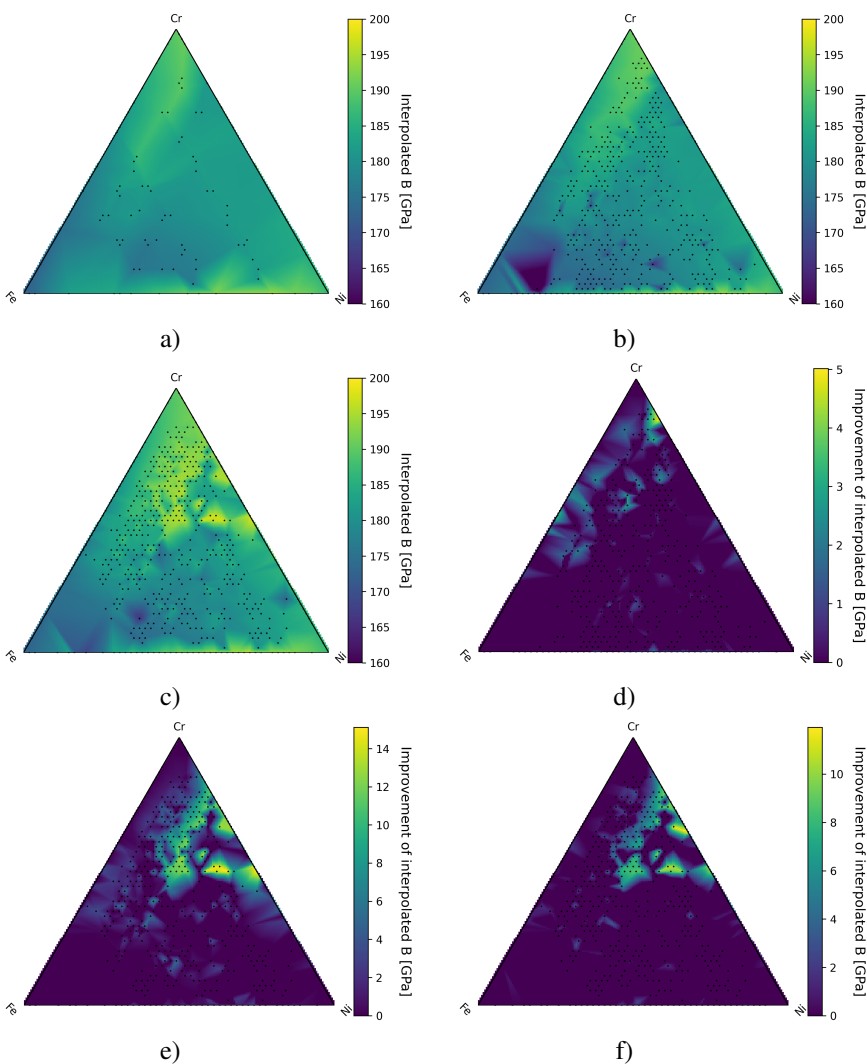

Figure 1: Plots a-c: interpolation of best-by-composition bulk moduli (denoted as B) from various portions of data: examples from the test set (a); examples from the test set after addition of examples generated by: P-CDVAE (b) or Aug-P-CDVAE (c). Plots d-f: improvements of interpolated bulk moduli due to addition of structures yielded by P-CDVAE to test set (d), and Aug-P-CDVAE to test (e) and augmented train (f) set examples.