# OpenReview forum: "Compositional Search of Stable Crystalline Structures in Multi-Component Alloys Using Generative Diffusion Models"
_ICLR.cc/2024/Conference — Submitted to ICLR 2024_

### Official Review · Reviewer_uusi · 2023-11-02

**Soundness:** 2 fair
**Presentation:** 1 poor
**Contribution:** 2 fair
**Rating:** 3
**Confidence:** 2

**Summary:**

This submission seems to use a generative model to search for new HEA structures.

**Strengths:**

The problem seems to be new and significant.

**Weaknesses:**

The paper is too hard to follow. I'd like the make clear that I am trained in machine learning and have a weak background in chemistry. However, I have no problem reading the paper "Crystal Diffusion Variational Autoencoder for Periodic Material Generation".

In this paper, I understand the goal is to search for new HEA structures. But I don't understand how this is achieved by the work. I think the submission does not have a clear problem formuation, e.g. what is the structure of one training sample?

 Here is a list of detailed questions.

1. In "Dataset creation with MD", what is one data point in the dataset? Is it a graph with node features? Could you give a clear definition of the sample space?

2. "Diffusion autoencoders are trained in a self-supervised manner, by removing the artificially induced noise of varying magnitudes from the data." What is the input to the model? How is the self-supervision task formed?

3. The training of "P-CDVAE": is the phase an observed variable so that you can trained the denoising model to predict the phase?

4. "Local search for data augmentation": the data augmentation seems to be different from data augmentation in machine learning. Here the local search is used to optimize some property of structure, instead of increasing training data.

5. "In order to cover the random ordering of the atoms in a specific point in the composition space," I don't see why node ordering should play a role here.

**Questions:**

Please check my questions above.

---

> ### Author Response · Authors · 2023-11-29
>
> We thank the referee for the feedback provided to the project.
>
> We understand the fact that the presentation aspect of the work is not thoroughly designed for the machine learning community. To solve this, we are planning to do a major revision on the text. Many aspects of the presented framework are rooted deeply in materials science - for example, when utilizing Molecular Dynamics simulations, the stability of structures is a major concern. We will be more careful about the extent to which we describe such matters - whether they should constitute the main text of the article, or serve as confirmation within the appendix - in order to better suit the area of interest with our next submission.
> In the following, we are answering to the referee’s questions:
>
> 1.  One data point is a crystal graph: apart from the node features (atom types), it is equipped with lattice vectors - those are three vectors along which the nodes repeat periodically. Lattice vectors can be seen as the features of the entire structure (graph), similarly to other input parameters: the value of the property (bulk modulus), and - for P-CDVAE - also a label that describes the phase (whether it’s an FCC or BCC structure). The output of the model has the same format and while theoretically, the sample space is not constrained to graphs of a certain size, in practice, all training examples were composed of 54 atoms, just like the training examples.
>
> 2. The input to the model is a crystal structure. The model is trained on the task of data denoising/reconstruction - after the structure is encoded, Gaussian noise is applied to the representation. The model learns to reconstruct the input graph and a loss function is applied based on the errors in reconstructed, which can be seen as a self-supervised task: in absence of external labels, a reconstruction task is performed based solely on the input data.
>
> 3. The crystal phase of the training examples (FCC or BCC) was straightforward to assess - the results of MD simulations aligned perfectly with the composition-to-phase relation visualized in FIG. 2, based on (Wu et al., 2017). Therefore, the structures in the dataset were annotated with corresponding crystal phase labels based purely on the composition. To assess the phase of the output structures, OVITO software was used. A loss function was applied to the model’s fully connected subnetwork to enforce correct prior phase prediction (against the phase labels). The training didn’t involve the loss function that explicitly punishes the mismatch between the phase of the emergent output structure and the label (or uses the phase of the output structure in any other way), but it did involve the coordinate loss. In that sense, the observation of the phase of output structure was not a part of the machine learning pipeline and was only conducted to analyze the results (fig. 4)
>
> 4. The goal of the operation was indeed to enrich the dataset with optimized structures that cannot be simply seen as sampled from uniform distribution (in terms of initial conditions, that then converge to locally stable spatial distribution). In hindsight, we admit that “augmentation” strikes as an inadequate name for this procedure - given that the goal of augmentation is usually to add examples to the dataset through simpler operations in order to account for simple feats (e.g. symmetries) of already existing cases.
>
> 5. We’re sorry to admit that this is one of the very unfortunate phrases that make the ideas behind the methodology obscure. A paraphrase that better conveys what was meant here is: “In order to cover more arbitrary orderings of atoms that are possible for a given atomic composition”.

---

### Official Review · Reviewer_1Sk8 · 2023-11-02

**Soundness:** 2 fair
**Presentation:** 1 poor
**Contribution:** 2 fair
**Rating:** 3
**Confidence:** 3

**Summary:**

This paper addresses the problem of discovering new, stable high-entropy alloys with generative machine learning. In particular, the paper proposes a modification of the Crystal Diffusion Variational Autoencoder (CDVAE) to be able to classify the phases of the ternary allow NiFeCr. Besides that, the paper contributes a data set for the aforementioned alloy and a method for augmenting the data set.

**Strengths:**

One of the positive aspects I see in this paper is that it tackles a problems that remains mostly unexplored in the machine learning for materials community, namely the discovery of high-entropy alloys with generative models. Another strength is the contribution of a data set for this problem, which the authors mention that would be made available upon acceptance (specifically, the data set creation workflow). The data set includes DFT and MD simulations.

**Weaknesses:**

I have a few important concerns that I would like to discuss.

First of all, in my opinion the presentation and clarity of the manuscript could be largely improved. One challenge I faced throughout the paper is the difficulty to understand important details of the problem, methods and result. I believe that an important reason for this difficulty is the extensive used of materials or physics jargon. Even though I can safely say that my materials and physics background is stronger than the average machine learning researcher's, I had significant difficulty in following some sections. Therefore, I suspect that it would also be hard for most of the NeurIPS audience. To point at some examples, one is Section 3.4 Reconstruction Scores, which refers to multiple methods that will probably be unknown for the majority of the machine learning community, including many of those working in materials-related applications. Section 3.1 was also particularly difficult to parse for me. More generally, I think that especially Section 3 could be further adapted to the NeurIPS audience. As another example transversal to the entire paper, I would note that the concept of "phase" plays a central role throughout the paper but it is nowhere defined. While the specific meaning in materials discovery or specifically in the domain of high-entropy alloys will obvious to the people in the field, this is a word with multiple meanings depending on the domain, even in different physics subdomains. I would recommend to explain as much as possible such concepts taking into account the target audience of the conference.

There are other aspects of the presentation that could be improved, in my humble opinion. For instance, the use of the figures could better support the text and ideally be self-explanatory. Figure 1 is not referred to in the text and the three-letter caption does not help a lot in its interpretation. Figure 2 also has a very short caption and I have not been able to make sufficient sense out of it, even after carefully reading the text. If I understand correctly, Algorithm 1 in page 5 refers to the property optimisation described in Section 3.3. However, the caption is merely a title and the algorithm contains a multitude of variables that are barely described in the text. Incidentally, I would strongly recommend to use the LaTeX mathematical mode to write mathematical variables in the text (see, for instance, the second-to-last paragraph in Section 3.3).

Generally, one aspect of the presentation that could potentially be improved in multiple sections is that I found it difficult to understand what are the important pieces of information and what are less important details. I would recommend trying to devise a way to help the reader, of the machine learning community, understand the contributions of the paper, both with a clear structure and descriptions as well as with the help of diagrams.

I found the discussion of the related work rather shallow. I think the paper effectively identifies relevant articles from the literature but falls short at putting them in context and explaining their relevance for the present manuscript.

Regarding the contributions, while I acknowledge that the domain is mostly unexplored, I would also note that the proposed method is a slight modification of the existing CDVAE and that the data set and experimental evaluation is limited to a specific alloy (NiFeCr). Therefore, it is uncertain whether the proposed framework will be applicable to other alloys and whether future research can easily build upon this work.

**Questions:**

My only question to the authors is whether they plan to make the data set available. The paper mentions that the "workflow" will be made available but it is unclear to me whether that will be sufficient for the community to follow up the work.

---

> ### Author Response · Authors · 2023-11-29
>
> We express our gratitude to the referee for acknowledging the novelty of the work in the high entropy alloy discovery field.
>
> We also understand the referee’s concerns regarding the presentation aspect of the work. When correcting the paper, our most important focus will be to largely clarify the description of our methodology, mostly the model architecture and the improvements we’ve included in the source model. We have indeed used too much materials science and physics jargon. To address this, we will focus on explaining these concepts, such as section 3 of the paper as the referee mentioned, in a more comprehensive manner for a machine learning scientist reader. In addition, we will provide detailed information, together with clear examples, in the supplementary materials section. This includes defining the concept of “Phase” in this context. The other concepts that we recognize need further explanation and clarification are for example: BCC, FCC, CALPHAD, MD, DFT, MEAM and etc. All the mentioned presentation issues will be carefully taken care of in the new version of the paper.
>
> We also agree with the referee on the issues regarding the figure captions and the “local-search” algorithm caption and description. In the new version, we will revise all the figure captions by adding more details about that specific figure. Also, we will add a detailed explanation of the algorithm with all the variables explained in the text. The LaTeX mathematical mode will be used for the equation as well. We will also work on the architecture of the paper, in a way that the readers have an easy time going through the paper.
>
> The “related work” section will be indeed revised thoroughly, explaining how this work is connected with the aforementioned literature.
>
>  Regarding the generalization of the experiment, we need to mention that the choice of experimental data was mainly dictated by the availability of high-quality interatomic potentials, necessary for embedding the computationally efficient MD simulations within the workflow. This raised concerns about the generalization capability of the approach. While MD poses some limitations to employing the workflow to arbitrary kinds of structures, we will put more focus on highlighting the possibility of applying the neural model itself to other data. We will contrast the results with the baseline CDVAE model on existing datasets, such as the Carbon dataset that posed a challenge to the original architecture.
>
> Finally, we are planning to make the dataset creation workflow available when the work is revised thoroughly. This will certainly make sense when the changes to the CDVAE architecture is reliable enough so that “any'' kind of dataset gives valuable results for high entropy alloy discovery.

---

### Official Review · Reviewer_dfH7 · 2023-11-23

**Soundness:** 2 fair
**Presentation:** 2 fair
**Contribution:** 2 fair
**Rating:** 3
**Confidence:** 3

**Summary:**

The authors propose a method and workflow for the generation of stable crystal structures, specifically targeting high entropy alloys.
The method is based on a cystal diffusion variational autoencoder (CDVAE) with an added classification model that predicts the crystal structure (FCC or BCC) based on the latent space representation.

**Strengths:**

From a modeling point of view, the biggest contribution of this work is the addition of a phase prediction network, that predicts whether the local structure around an atom is FCC or BCC. According to the authors' CP G-distance and qualitative visualisations this improves the structural integrity of the generated crystals structures.

**Weaknesses:**

Overall the paper is difficult to follow for machine learning specialists (main ICLR audience), because the main discussion points are with respect to the properties of the generated structures and less about the design choices regarding the model and the evaluation of the proposed improvement.
It is unclear whether the "local search for data augmentation" is used for expanding the training set or it is used to jump out of local minima during the langevin dynamics optimization of structures.
The authors introduce a local reconstruction score. It is not clear to me whether this metric is used as a replacement for the
reconstruction error of the CDVAE during training or if it is only used as a final evaluation metric. And why is it important to only score the local structure. And if it is better than the usual reconstuction error of the VAE, why is it not used as the cost function for training? In 4.4 the authors write that the model is able to find bulk modulus that are similar to the values found during local search. How much faster is the search using the model vs the local search that doesn't use machine learning? Did the model use the local search data for training? If that is the case we don't really gain anything from using the model.

Overall the presented results might be good and sound, but the framing of the paper and each experiment is not clear enough
to make a judgement about the soundness of the results.

**Questions:**

Do you use different cell sizes (number of atoms) for training and testing? Does the model generalize to different number of atoms?
How big is the training set?
During training do you use the output of your crystal phase classifier as input to the decoder or do you feed the label directly into the decoder instead of predicting it?

---

> ### Author Response · Authors · 2023-12-02
>
> We Thank the referee for the valuable feedback.
>
> The G-vector based metrics are not used during the training of the model and are only utilized for final evaluation. The choice of scoring the local structure was made based on our primary assumptions regarding the task itself, as well as the way we expect CDVAE to act in our setting:
> Firstly, we assume that the training step of the diffusion model is indeed aimed at the exact reconstruction of the training example, not just the local patterns. That being said, we expect the local patterns to have a large influence on the bulk modulus: they introduce distortions in the crystal lattice, influencing the structure’s mechanical properties. To provide sufficient freedom for the model to form diverse patterns, we provide a relatively large supercell of 54 atoms. When vaguely saying “patterns”, we mean atomic clusters of various sizes: this could be, for example, a bunch of atoms with a bigger radius surrounding one with a smaller radius.
> Within the supercell, those patterns can appear shifted by symmetric transformations (translation/rotation), which doesn’t concern us at all, or separate clusters of atoms could be reconstructed correctly, but have different relative positions to each other, which we consider an imperfection, but not the most critical one.
>
> To justify why we don't expect the perfect match overall, let's assume the following scenario: the model's fully-connected atomic composition submodule, which is queried before Langevin dynamics start, predicts that the structure consists of 52 atoms of one kind and 2 atoms of another kind. For training, noise is applied to the positions of those atoms - therefore, we ideally expect them to end up in the same place as before applying the noise. In sampling, the atoms will start in randomly chosen positions, to then form a regular lattice. We don't control for the exact placement of the two atoms of a certain kind, but we want their relative positions to be the same.
> One of the ways to check if this goal was achieved is to utilize tools to match two crystal structures in order to account for rotational or translational symmetries before evaluating the pattern. In the experimental setting of our choice, due to imperfect reconstructions by the model, as well as overall size of the matched structures, the usage of those tools proved to be very problematic. We opted for the alternative approach - producing the embeddings of each atom, based on the positions (distances and angles) and atom types of their neighbours. This descriptor-based approach also gives a measure of fulfilling our goal - in order to assume the perfect reconstruction, the relative positions/atom types of their neighbours have to be the same - only then, the atom’s embedding will be an exact match to the ground truth.
>
> The model in question (one used for the final generation of structures) did use the local search data for training. While it found structures with highest bulk modulus than the best examples seen during training, the magnitude of improvement upon this baseline is indeed small. The inclusion of the examples produced with local search had significant impact on the accuracy of reconstruction of the structures - one can hypothesize that the initially produces dataset, which consisted only of randomly-initialized structures, was not the best domain for a diffusion model, which is expected to iteratively distinguish between the random noise and the underlying signal. Local search certainly helped in the regard of including information that is certainly “non-random”. As the degree of feature improvement may be seen as insufficient, we acknowledge that further tests should be done to better distinguish between what was seen during the training and what was not.
>
> Both training and testing datasets consist only of 54-atom structures. Based on the training data provided, the model generalizes to different (and varying) numbers of atoms. 1150 examples were created for the database. We use the output of our crystal phase classifier both during the training and sampling. In practice, for the studied case, the distinction between different crystal phases is a trivial matter: it can be reliably predicted based solely on the summary formula (nevertheless, perhaps not as easy for the denoising GNN, which, for this application, is not equipped with any pooling layer or mechanism that could correctly view the structure “as a whole”; apart from the raw structure encoding passed to each atom). One can assume the difference between the two approaches mentioned in the question would be marginal on NiFeCr dataset, however, for the cases where this prediction is more intricate, the distinction between them could be much more relevant.

---

### Meta-Review · Area_Chair_n9qe · 2023-12-08

**Metareview:**

This paper proposes an extension to the Crystal Diffusion Variational Autoencoder (CDVAE).

All the reviewers agree that the paper is very hard to follow and not as it stands adequate for the ICLR audience. Rejection is therefore recommended. The authors are encouraged to use the valuable feedback from the referees, update the paper and submit it for publication elsewhere.

**Justification For Why Not Higher Score:**

Unnecessarily hard to follow. It is extra important for a paper which is presenting an application at the periphery of what machine learners normally work with.

**Justification For Why Not Lower Score:**

None.

---

### Decision · Program_Chairs · 2024-01-16

Reject